# Accelerating Anchors via Specialization and Feature Transformation

## Abstract

Anchors is a popular local model-agnostic explanation technique whose applicability is limited by its computational inefficiency. To address this limitation, we propose a pre-training-based approach to accelerate Anchors without compromising the explanation quality. Our approach leverages the iterative nature of Anchors' algorithm which gradually refines an explanation until it is precise enough for a given input by providing a general explanation that is obtained through pre-training as Anchors' initial explanation. Specifically, we develop a two-step rule transformation process: the horizontal transformation adapts a pre-trained explanation to the current input by replacing features, and the vertical transformation refines the general explanation until it is precise enough for the input. We evaluate our method across tabular, text, and image datasets, demonstrating that it significantly reduces explanation generation time while maintaining fidelity and interpretability, thereby enabling the practical adoption of Anchors in time-sensitive applications.

## 1 Introduction

Anchors Ribeiro et al. (2018) is a popular local model-agnostic machine learning explanation technique, which generates rules that form a sufficient condition to explain why a model makes a certain prediction for a given input. It is local in the sense that it explains a machine learning model's behavior around a particular input, which makes it able to scale to complex models in critical domains such as healthcare and finance, where understanding individual predictions is vital for decision-making and validation. It is model-agnostic in the sense that it does not exploit internal design of a model, which makes it applicable to a wide range of machine learning models. These distinct features make Anchors adopted in many important applications Arya et al. (2022) Sarkar & Bala (2022), such as explaining a triage-prediction system for COVID-19 to enable knowledge discovery about patient risk factors Khanna et al. (2023), and analyzing Existing Vegetation Type to uncover insights for ecological patterns and land management Ganji & Lin (2023). However, a significant bottleneck hinders its widespread application: the inherent computational inefficiency. The interpretation time for a single input of Anchors can last for several hours in certain scenarios, impeding the practical deployment of Anchors in real-time applications where prompt explanations are crucial.

In this paper, we attempt to find a method to improve the computational efficiency of Anchors without compromising the quality of the generated explanations. To achieve this, we make two key observations: 1) most of Anchors' computation time is spent in drawing samples that are variants of the input to explain upon, and 2) Anchors draw samples in an iterative manner to build the explanation incrementally that becomes more and more specific to the current input.

Based on these observations, we propose a method to improve the Anchors' efficiency: we pre-train explanations for representative pre-training inputs in advance, and then during online computation, for an online input, start the iterative computation process with a pre-trained explanation that is obtained from a similar pre-training input, thereby reducing the time spent in drawing samples.

However, our method faces two major challenges: 1) The decision boundaries of the model being explained in these neighborhoods may differ. Therefore, explanations that are effective for pre-training inputs may not be effective for online inputs. 2) For two similar but not identical inputs, their features are not completely consistent. Consider the example in Table 1, where pre-trained sample $x_1$ and input sample $x_2$ do not share the same features, although their explanations are very similar.

| Input sample $\boldsymbol{x}$ | Generated explanation $R_x$ | $f(\boldsymbol{x})$ |
|---|---|---|
| $\boldsymbol{x}_1$ = This is the best movie that I've ever seen. | ("best") | positive |
| $\boldsymbol{x}_2$ = He was really nice today and helped me a lot. | ("nice") | positive |

Table 1: The sampling process with different features.

Since Anchors generate explanations by using features in its rules, explanations applicable to pre-trained inputs may not necessarily be applicable to new inputs. Consequently, the rules generated from $\boldsymbol{x}_1$ cannot initiate explanations for $\boldsymbol{x}_2$.

To address these two challenges, we propose a method based on transforming rules to accelerate the generation of explanations. We propose two forms of rule transformation: one is horizontal, transforming a rule for one input into a rule for another input by substituting the features; the other is vertical, transforming a general rule (rule with high coverage and low precision) into a specific rule (rule with low coverage and high precision) by strengthening it.

We conduct the experiments to demonstrate that our method achieves acceleration ratios of 271%, 221%, and 181% on various tasks across tabular, text, and image datasets, respectively. Moreover, with sufficient pre-training, it can achieve the fidelity consistent with that of Anchors.

## 2 BACKGROUND

This chapter provides the necessary background knowledge that is integral to understanding our approach. We first describe the form of Anchors' explanations and then its underlying algorithm.

### 2.1 REPRESENTATION OF ANCHORS' EXPLANATIONS

We first introduce several basic terms used in our framework. Let $\mathbb{X}$ denote the input space and $\mathbb{Y}$ the label space. A target model is a function $f : \mathbb{X} \to \mathbb{Y}$. An input instance $\boldsymbol{x} \in \mathbb{X}$ is represented as a tuple $\boldsymbol{x} = (\boldsymbol{x}_1, \boldsymbol{x}_2, ..., \boldsymbol{x}_n)$, where each $\boldsymbol{x}_i$ denotes the $i$-th feature and $n$ is the total number of features. A predicate is a function that maps an input $\boldsymbol{x}$ into a binary value, i.e., $p : \mathbb{X} \to \{0, 1\}$. Specifically, the predicates limit a feature to a specific value, denoted as $p_{i,v}(\boldsymbol{x}) := \mathbf{1}[\boldsymbol{x}_i = v]$, where $\mathbf{1}[\cdot]$ is the indicator function. We now turn to the formal mathematical constructs that define the behavior and objective of Anchors' algorithm.

**Definition 2.1** (Rule). For an input $\boldsymbol{x}$ of $n$ features, a rule $r \in \mathbb{R}_{\boldsymbol{x}}$ is a conjunction of predicates: $r := p_{a_1,v_1} \wedge p_{a_2,v_2} \wedge \cdots \wedge p_{a_m,v_m}, m \le n$.

Similar to a predicate, we treat a rule as a function, i.e., $r(\boldsymbol{x}) = 1$ when $\forall i \in [1, m].p_{a_i,v_i}(\boldsymbol{x}) = 1$. In addition, we use $\mathbb{R}$ to denote a set of rules. Since our discussion does not involve the set of real numbers, all occurrences of $\mathbb{R}$ hereafter refer to the rule set.

**Definition 2.2** (Perturbation Distribution). Given $\boldsymbol{x} \in \mathbb{X}$, let $D_{\boldsymbol{x}}(\cdot)$ denote a probability distribution over perturbed instances $\boldsymbol{z} \in \mathbb{X}$ near $\boldsymbol{x}$. The conditional distribution $D_{\boldsymbol{x}}(\cdot \mid r)$ is defined as:

$$D_{\boldsymbol{x}}(\boldsymbol{z} \mid r) := D_{\boldsymbol{x}}(\boldsymbol{z}) \cdot \mathbf{1}[r(\boldsymbol{z}) = 1],$$

Next, we define the two most important performance metrics of Anchors as follows:

**Definition 2.3** (Precision). Given a rule $r$, the precision is:

$$\text{Precision}(r) := \mathbb{E}_{\boldsymbol{z} \sim D_{\boldsymbol{x}}(\cdot \mid r)} \left[ \mathbf{1}[f(\boldsymbol{z}) = f(\boldsymbol{x})] \right].$$

**Definition 2.4** (Coverage). The coverage of rule $r$ is:

$$\text{Coverage}(r) := \mathbb{E}_{\boldsymbol{z} \sim D_{\boldsymbol{x}}(\cdot)} \left[ \mathbf{1}[r(\boldsymbol{z}) = 1] \right].$$

Precision measures how accurate the explanation approximates the target model in a given input set (i.e., ones satisfying the explanation rule), while coverage measures how large the set is. They together measure how faithful the explanation is. Next, we formally define the output of Anchors.

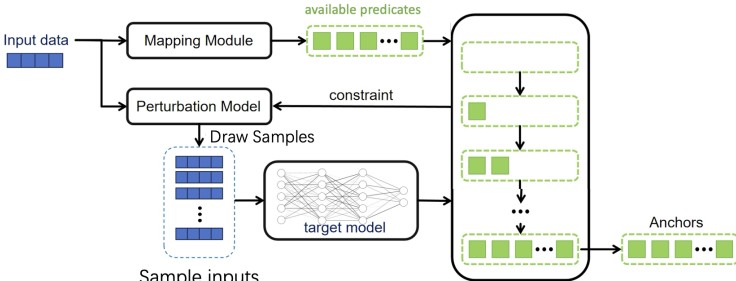

Figure 1: The overall workflow of Anchors.

**Definition 2.5** (Anchors Output). Let $\tau \in [0, 1]$ and $\delta \in [0, 1]$ be thresholds. The Anchors explanation rule for model $f$ and input $\boldsymbol{x}$ is a rule covering $\boldsymbol{x}$, meeting the precision threshold $\tau$ with high probability $\delta$ while maximizing the coverage:

$$r_{\boldsymbol{x}} \in \text{argmax}_{r \in R_p} Coverage(r)$$

where $R_p = \{\hat{r} \mid r(\boldsymbol{x}) = 1 \wedge P(Precision(\hat{r}) \geq \tau) \geq 1 - \delta \wedge \hat{r} \in \mathbb{R}_{\boldsymbol{x}}\}$

### 2.2 THE ALGORITHM OF ANCHORS

We now present the Anchors' algorithm, since our method involves algorithmic changes to Anchors.

As shown in Figure 1, the Anchors algorithm mainly consists of the following three steps:

- It generates a set of available predicates $\mathbb{P}$ based on the input $\boldsymbol{x}$.
- It adds each predicate $p_{i,v}$ from the predicates set $\mathbb{P}$ that has not yet appeared in a rule named $r$ (which initially contains no predicates) to form the candidate rule set $\mathbb{R}_{next} = \{r \wedge p | p \in \mathbb{P} - r\}$. It uses the KL-LUCB algorithm Kaufmann & Kalyanakrishnan (2013) to determine the sample number $M$, and feeds $r$, $M$, and $\boldsymbol{x}$ into the perturbation model to generate the neighborhood $D_{\boldsymbol{x}}$ of the input $\boldsymbol{x}$. Then it calculates *coverage* and *precision* of all the rules in rule set $\mathbb{R}_{next}$ using $D_{\boldsymbol{x}}$.
- If the precision of any rule is no less than $\tau$, it returns a rule that has the maximum *coverage* among all rules satisfying the precision requirement. Otherwise, it sets $r$ to be the rule with the highest precision and continues to Step 2.

It can be seen that after each iteration, one predicate is added to the rule $r$. As the number of predicates in $r$ increases, the number of inputs it can cover will decrease, while the probability that the prediction results of the covered inputs are the same as the original input's will increase. In other words, during the iteration, the *coverage* will gradually decrease while the *precision* will gradually increase, and the rule $r$ will gradually transform from a "general" explanation to a "specific" explanation for a particular input.

## 3 OUR APPROACH

To reduce Anchors' online computational cost, our method first pre-trains on a set of representative inputs by pre-computing their explanations. During online explanation, instead of running Anchors from scratch, our method finds the most similar representative input and adapts its explanation through two transformation steps:

- A **horizontal transformation**, which modifies the pre-trained rule to match the feature space of the new input.
- A **vertical transformation**, which incrementally refines this adapted rule to meet a required precision threshold.

These steps allow us to significantly reduce the number of samples required during online explanation. This section provides a detailed explanation of each component of our approach. We begin with a high-level overview to motivate and illustrate the transformations.

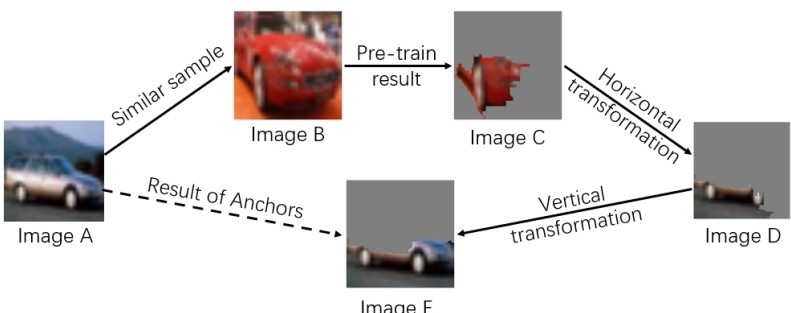

Figure 2: An example of our workflow for image.

---

**Algorithm 1** Pre-training process

---

**Input**: The dataset $\mathbb{X}$ and the model to explain $f$
**Parameter**: The number of pre-training inputs $N$, the precision threshold $\tau$ and the probability threshold $\delta$
**Output**: The pre-training result $\mathbb{R}_{pre}$

1: $\mathbb{X}_{pre-train} := \text{K-means}(\mathbb{X}, N)$
2: $\mathbb{R}_{pre} := [\,]$
3: **for all** $\boldsymbol{x} \in \mathbb{X}_{pre-train}$ **do**
4:     $r := Anchors(f, \boldsymbol{x}, \tau, \delta)$
5:     $\mathbb{R}_{pre}.add(r)$
6: return $\mathbb{X}_{pre-train}, \mathbb{R}_{pre}$

---

### 3.1 OVERVIEW

We use the image-classification example in Figure 2 to show our method's workflow and how horizontal and vertical transformations operate. Anchors explain images as sets of superpixels that serve as sufficient conditions: if all are present, the model likely repeats its original prediction.

Suppose Image A is classified as a car. Our approach retrieves a similar image, Image B, with a pre-computed anchor (Image C), often representing a generic feature like wheels. To ensure generality, this pre-trained anchor uses few superpixels, giving high coverage but low precision.

Because Image B's anchor differs in pixel-level representation, it cannot be applied directly to Image A. We first apply a *horizontal transformation* to adapt it, replacing its superpixels with visually similar ones in Image A to produce a new anchor (Image D) that covers relevant parts but still lacks precision. Next, a *vertical transformation* refines the explanation by adding superpixels until the desired precision is reached, yielding a final anchor (Image E) that matches the original Anchors output with fewer samples and faster computation.

The following subsections describe our approach's offline pre-training and online refinement phases.

### 3.2 OFFLINE PRE-TRAINING

Algorithm 1 outlines the pre-training phase, which takes an input set $\mathbb{X}$ and a model $f$ as input and outputs a set of rules. The parameter $N$ specifies the number of inputs to generate rules on. The other parameter $\tau$ specifies the precision requirement for Anchors during pre-training, which balances the precision and coverage of the generated rules. The higher $\tau$ is, the more precision and the lower coverage the generated explanation rule has. To generate general rules, the pre-training phase uses a lower precision threshold compared to the online explanation phase.

In Line 1, the algorithm uses the K-means clustering algorithm Krishna & Narasimha Murty (1999) to identify $N$ clusters and selects their centroids as representative inputs (stored in $\mathbb{X}_{pre-train}$). In Line 3-6, the algorithm iterates through each input in $\mathbb{X}_{pre-train}$ and applies Anchors with specified precision threshold $\tau$ to generate explanation rules. Finally, the algorithm returns the pre-training input set and all the corresponding generated explanation rules in line 7.

---

**Algorithm 2** Refinement Process

---

**Input:** The input $\boldsymbol{x} = (\boldsymbol{x}_1, \boldsymbol{x}_2, ..., \boldsymbol{x}_m)$, the model to explain $f$, the pre-training dataset $\mathbb{X}_{pre-train}$ and the pre-training results $\mathbb{R}$
**Parameter**: The precision threshold $\tau$ and the probability threshold $\delta$
**Output**: The explanation $r$

1:  $\hat{\boldsymbol{x}}_{similar} := ChooseAny(arg\,min_{\hat{\boldsymbol{x}}_i \in \mathbb{X}_{pre-train}} Dist(\boldsymbol{x}, \hat{\boldsymbol{x}}_i))$
2:  $r_1 := \mathbb{R}_{pre}[\mathbb{X}_{pre-train}.index\,of(\hat{\boldsymbol{x}}_{similar})]$
3:  $\# \, Horizontal \, Transformation$
4:  $r_2 := \emptyset$
5:  **for** $p_{i,v}$ **in** $r_1$ **do**
6:     $similar\_feature := \mathbf{None}$
7:     **for** $j = 1$ **to** m **do**
8:        **if** $Dist(v, \boldsymbol{x}_j) \leq Dist(v, similar\_feature)$ **then**
9:           $similar\_feature := \boldsymbol{x}_j$
10:    $r_2 := r_2 \wedge \{p_{i,similar\_feature}\}$
11: $\# \, Vertical \, Transformation$
12: $r = r_2$
13: **repeat**
14:    $\mathbb{R}_{next} := \emptyset$
15:    **for** $i = 1$ **to** m **do**
16:       **if** $p_{i,\boldsymbol{x}_i} \notin r$ **then**
17:          $\mathbb{R}_{next} := \mathbb{R}_{next} \cup \{r \wedge \{p_{i,\boldsymbol{x}_i}\}\}$
18:    $r := ChooseAny(arg\,max_{r_i \in \mathbb{R}_{next}}(precision_{\boldsymbol{x},f}(r_i)))$
19: **until** $P(precision_{\boldsymbol{x},f}(r) \geq \tau) \geq 1 - \delta$
20: return $ChooseAny(arg\,max_{r_i \in \mathbb{R}_{next}}(coverage_{\boldsymbol{x},f}(r_i)))$

---

## 3.3  ONLINE REFINEMENT

Algorithm 2 outlines the online refinement phase of our approach. It takes an input $\boldsymbol{x}$, a model $f$, the pre-training dataset $\mathbb{X}_{pre-train}$, and the corresponding pre-trained rules $\mathbb{R}_{pre}$ as input. Additionally, it includes a parameter $\tau$ that controls the precision of the Anchors explanation, as in Algorithm 1. Here, $\tau$ is set higher than in Algorithm 1 to return a more specific rule to the user. The output of the Algorithm 2 is consistent with Anchors, producing an explanation $r$ for the input $\boldsymbol{x}$ and the model $f$. Next, we discuss the details of Algorithm 2 and provide formal definitions for HT and VT.

**Obtaining a similar input.** In Line 1, the algorithm obtains the pre-training input $\hat{\boldsymbol{x}}_{similar}$ that is most similar to the input $\boldsymbol{x}$. In this process, we map the input $\boldsymbol{x}$ and the pre-trained input set $\mathbb{X}$ to the same embedding space. Based on the distance in the embedding space, we identify the pre-training input that is closest to $\boldsymbol{x}$ and treat it as the most similar pre-training input, $\hat{\boldsymbol{x}}_{similar}$. Then in Line 2, the algorithm retrieves the pre-trained rule $r_1$ of $\hat{\boldsymbol{x}}_{similair}$ produced in Algorithm 1.

**Horizontal transformation.** From Line 4 to Line 13 is the horizontal transformation (HT). The HT maps the predicates from the pre-trained rules onto the predicates formed by the features in the online input that are most similar to them. We first give the mathmatical definition of HT, and then introduce its implementation:

**Definition 3.1** (Horizontal Transformation). Let $r_1 = p_{a_1,\boldsymbol{x}_{a_1}} \wedge p_{a_2,\boldsymbol{x}_{a_2}} \wedge \cdots \wedge p_{a_n,\boldsymbol{x}_{a_n}}$ be a rule derived from a pre-trained input $\boldsymbol{x}$. The function HT $: \mathbb{R}_{\boldsymbol{x}} \to \mathbb{R}_{\boldsymbol{x}'}$ maps $r_1$ to a new rule $r_2 = p_{b_1,\boldsymbol{x}'_{b_1}} \wedge p_{b_2,\boldsymbol{x}'_{b_2}} \wedge \cdots \wedge p_{b_m,\boldsymbol{x}'_{b_m}}$ for the online input $\boldsymbol{x}'$, where $m \leq n$ and each predicate $p_{i,\boldsymbol{x}_i}$ in $r_1$ is transformed into a predicate $p_{j,\boldsymbol{x}'_j}$ in $r_2$ such that $\boldsymbol{x}'_j$ is the most similar feature to $\boldsymbol{x}_i$ based on the following criterion:

$$\text{Dist}(\boldsymbol{x}_i, \boldsymbol{x}'_j) \leq \text{Dist}(\boldsymbol{x}_i, \boldsymbol{x}'_k), \quad \forall k \in \{1, \ldots, m\}$$

The function $\text{Dist}(\cdot, \cdot)$ represents a distance metric defined in the perturbation space. For tabular data, it represents the absolute value of the difference between two numbers. For text data, it represents the distance between two words in the semantic space generated by a fine-tuned BERT Devlin et al. (2018). For image data, it represents the distance between the vectors of two superpixels after embedding via Resnet50 He et al. (2016).

Then we introduce the detailed steps of HT: In Line 5, the algorithm enumerates each predicate $p_{i,v}$ in the pre-training result $r$. From Line 7 to Line 11, the algorithm identify the most similar feature in the input $x_j$ based on the value $v$ in $p_{i,v}$. At last, in Line 12, the algorithm adds the predicate $p_{i,similar\_feature}$ to the rule $r_2$.

**Vertical transformation.** From Line 15 to Line 25 is the vertical transformation (VT). This process refines a general rule—which typically has high coverage but low precision—into a more specific rule with low coverage and high precision. The formal definition of VT is as follows:

**Definition 3.2** (Vertical Transformation). Let $\tau \in [0, 1]$, $\delta \in [0, 1]$ be thresholds. The function $VT : \mathbb{R}_x \to \mathbb{R}_x$ maps a basic rule $r_2$ into a rule covering $x$, implying $r_2$, meeting the precision threshold $\tau$ with high probability $\delta$ while maximizing the coverage:

$$r_x \in \text{argmax}_{r \in R_p} Coverage(r)$$

where $R_p = \{\hat{r} \mid r(x) = 1 \land P(Precision(\hat{r}) \geq \tau) \geq 1 - \delta \land \hat{r} \in \mathbb{R} \land \hat{r} \Rightarrow r_2\}$

This process works by continuously adding new predicates to the current rule $r$. At the start of each iteration (Line 17), the candidate rule set is initialized as empty. In Line 18, the algorithm enumerates all features of input $x_i$. Lines 19–20 add a new predicate $p_{i,x_i}$, not already in $r_2$, to form candidate rules. In Line 23, perturbation sampling selects the candidate with the highest precision as the new rule $r$ for the next iteration. This repeats until $Precision(r) \geq \tau$ with high probability, after which the rule with the highest $coverage$ is output (Line 25).

Thus, we align with Anchors' precision goals and ensure that explanations can be effectively adapted to specific contexts without extensive recalculations.

### 3.4 TIME COMPLEXITY ANALYSIS

Next, we present a theoretical analysis of our method's acceleration, deriving both the expected speedup and its bounds. The analysis focuses on the time complexity of generating an explanation for a single input. For the original Anchors method, assume the final explanation requires $k$ predicates, a single query to the target model has time complexity $O(T)$, the input $x$ has $n$ features, and the average complexity of one KL-LUCB run is $O(n \log(n) \log(n/\tau))$ Kaufmann et al. (2016). Thus, the overall average complexity of Anchors is $O(k \cdot n \log(n) \log(n/\tau) \cdot T)$.

Our method first extracts some predicates using HT, with negligible cost since it avoids sampling. VT then iteratively adds the remaining predicates while preserving Anchors' confidence using KL-LUCB and sampling. If HT provides $d$ predicates, VT adds $k - d$, giving a complexity of $O((k - d) \cdot n \log n \log(n/\tau) \cdot T)$. When $d < k$, the acceleration ratio is roughly $\frac{k}{k-d} \times 100\%$.

In the special case $k = d$, where HT provides all predicates, VT only performs a fast sampling step to ensure high confidence. This is the best-case scenario for KL-LUCB, with a complexity of $O(n \log(n/\delta) \cdot T)$, leading to an acceleration ratio of about $k \log(n) \times 100\%$. This estimate is approximate since KL-LUCB's runtime can vary. In experiments, some inputs reached $7700\%$ acceleration with $n = 13$ and $k = 4$.

Overall, optimization effectiveness depends on the number of predicates $d$ provided by HT, which in turn depends on whether pre-training samples sufficiently cover the input space. To explore this, we conduct experiments with varying pre-training input sizes $N$, described later.

## 4 EXPERIMENT

In this section, we aim to empirically evaluate the effectiveness of our method compared to the original Anchors algorithm. Our experiments are designed to answer the following key questions:

- **Efficiency Improvement:** To what extent does our method improve the efficiency of Anchors? Specifically, how much time and sampling cost are reduced?
- **Fidelity Perservation:** Does our accelerated method preserve the fidelity of the original Anchors explanations, in terms of precision and coverage?

To comprehensively address these questions, we evaluate our approach on three types of data—tabular, text, and image—using multiple machine learning models and datasets.

## 4.1 Experiment Setup

For tabular data, we selected revenue forecasting as the target task; for text data, we selected sentiment analysis as the target task; for image data, we selected image classification as the target task. The selection of the following datasets and models refers to the experiments of Anchors.

Across all tasks, we set $\tau = 0.8$ for pre-training and $\tau = 0.95$ for refinement, with $\delta = 0.05$. Since $N$ strongly affects optimization, we ran comparative experiments with different $N$ values. The training set was used for pre-training, and the test set for evaluation.

**Income Prediction.** The income prediction models take the numerical values of a person's multiple features (such as age, education level, race, etc.) as input and outputs whether their annual income will exceed \$50k or not, i.e. $f : X \rightarrow \{0, 1\}$, where $X := \bigcup_{i=0}^{i \leq k} F_i$ is the input domain, and $k$ represents the number of features, $F_i$ represents the value of i-th feature. We used the random forest (RF) Breiman (2001), gradient boosted trees (GBT) Chen & Guestrin (2016) and a 3-layers neural network (NN) Rumelhart et al. (1986) as models to explain. We used these models to predict the data of 12,345 individuals from the Adult dataset Becker & Kohavi (1996), and explained the local behavior of the models around each input item in tabular.

**Sentiment Analysis.** Sentiment analysis models take a text sequence as input and predict whether it is positive or negative, i.e. $f : X \rightarrow \{0, 1\}$, where $X := \bigcup_{i=1}^{\infty} W^i$ is the input domain, and $W$ is the vocabulary set. We used the random forest (RF) Breiman (2001) and the 7B version of Llama2.0 (Llama) Touvron et al. (2023) as the target models. We used these models to predict 12,520 comments from the RT-Polarity dataset Pang & Lee (2005), and explained the local behavior of the models around each input text.

**Image Classification.** Image classification models take an image as input and predict its category, i.e., $f : X \rightarrow \{0, 1, ..., m\}$, where $m$ is the number of categories and $X := \mathbb{R}^{3 \times h \times w}$ is the input domain, with $h$ and $w$ denoting image height and width. We used a pre-trained YOLOv8 Jocher et al. (2023) to classify images from an ImageNet Deng et al. (2009) subset. Because the full dataset is too large and Anchors is time-consuming, we randomly selected 500 test images for evaluation.

All experiments were conducted on a server with 4 high-performance GPUs (10,000+ CUDA cores and 24GB memory each) and 256GB system memory. The full run took about 300 hours.

## 4.2 Efficiency Improvement

### 4.2.1 Evaluation Metrics

To quantify the performance of our accelerated Anchors method, we measured the following metrics:

**Time Acceleration Ratio**: The time acceleration ratio of our method compared to the original Anchors, calculated as (Time taken by the original Anchors/ Time taken by our method).

Let $\text{time}(\cdot)$ denote computation time, $f$ represent the model to explain, $\tau$ denote the precision threshold, $X$ denote the dataset, and $X_{\text{pretrain}} \subseteq X$ represent the pre-trained input subset of $X$. Then, the **time acceleration ratio** ($R_{\text{acc}}$) can be expressed as:

$$R_{\text{acc}} = \frac{\sum_{x \in X} \text{time}(\text{Anchors}(f, x, \tau))}{\sum_{x \in X} \text{time}(\text{Accl}_{\text{Anchors}}(f, x, X_{\text{pretrain}}, \tau))}$$

Where $\text{Anchors}(f, x, \tau)$ refers to the baseline process for interpreting $f$ applied to $x$ with precision threshold $\tau$, and $\text{Accl}_{\text{Anchors}}(f, x, X_{\text{pretrain}}, \tau)$ represents our accelerated method.

In other words, the higher the $R_{acc}$, the more evident the optimization effect of our method.

**Sampling Reduction Ratio**: This metric is calculated as $1 - \frac{\text{Sampling count of our method}}{\text{Sampling count of Anchors}}$. It reduces the impact of model computational overhead, providing a clearer measure of acceleration. A lower ratio indicates stronger optimization.

### 4.2.2 Evaluation Result

Table 2 shows the average time acceleration and sampling reduction ratios across three tasks with varying numbers of pre-training explanations $N$.

| | Models to explain | N=100 | N=200 | N=500 | N=1000 | N=2000 |
|---|---|---|---|---|---|---|
| | | *Income Prediction* | | | | |
| Time Acceleration Ratio | RF | 158% ± 10% | 172% ± 11% | 175% ± 9% | 221% ± 13% | 271% ± 17% |
| | NN | 154% ± 7% | 181% ± 9% | 183% ± 7% | 189% ± 9% | 198% ± 12% |
| | GBT | 153% ± 4% | 156% ± 6% | 149% ± 5% | 167% ± 6% | 185% ± 8% |
| Sampling Reduction Ratio | RF | 41% ± 5% | 45% ± 5% | 47% ± 4% | 55% ± 6% | 64% ± 7% |
| | NN | 34% ± 3% | 40% ± 4% | 42% ± 4% | 51% ± 6% | 53% ± 6% |
| | GBT | 34% ± 3% | 36% ± 5% | 31% ± 3% | 42% ± 5% | 46% ± 6% |
| | | *Sentiment Analysis* | | | | |
| Time Acceleration Ratio | RF | 105% ± 6% | 113% ± 7% | 123% ± 8% | 158% ± 10% | 221% ± 13% |
| | Llama | 117% ± 7% | 136% ± 9% | 144% ± 10% | 158% ± 11% | 169% ± 13% |
| Sampling Reduction Ratio | RF | 6% ± 2% | 14% ± 3% | 20% ± 3% | 40% ± 5% | 57% ± 7% |
| | Llama | 13% ± 3% | 29% ± 4% | 34% ± 5% | 39% ± 5% | 42% ± 6% |
| | | *Image Classification* | | | | |
| | Models to explain | N=10 | N=20 | N=50 | N=100 | N=200 |
| Time Acceleration Ratio | YOLOv8 | 115% ± 9% | 138% ± 10% | 156% ± 11% | 163% ± 12% | 181% ± 13% |
| Sampling Reduction Ratio | YOLOv8 | 12% ± 3% | 29% ± 4% | 34% ± 5% | 42% ± 6% | 47% ± 7% |

Table 2: Acceleration effects of our approach on the three tasks (mean ± 95% confidence interval).

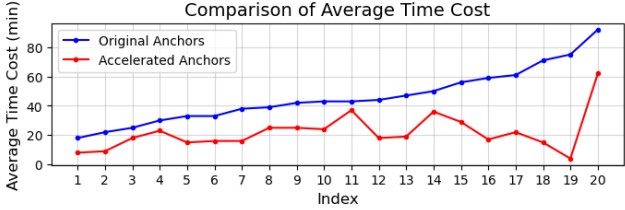

Figure 3: The absolute time cost comparison before and after acceleration.(Sentiment Analysis)

For the income prediction task, our method achieved the highest acceleration across all models. With $N = 2000$, the acceleration ratios for RF, NN, and GBT were 271%, 198%, and 185%, with sampling reductions of 64%, 53%, and 46%, respectively.

For more complex tasks like sentiment analysis and image classification, the acceleration remained effective. With $N = 2000$, the RF model achieved 221% acceleration, while Llama reached 169%. This gap arises because Llama's complex decision boundary requires more predicates, corresponding to a larger $k$ in our analysis. In the image classification task with $N = 200$, YOLOv8 achieved an acceleration ratio of 181%. Across all tasks, increasing $N$ consistently improved optimization, matching our theoretical predictions.

Figure 3 compares the absolute runtime of Anchors and our method on the sentiment classification task using the Llama model, which has the highest runtime in Anchors. We randomly selected 20 inputs and measured the average explanation time over 10 runs per input, sorting results by Anchors' runtime. Our method achieves greater speedup for inputs with higher time costs. However, inputs such as indices 4, 11, and 20 show limited optimization due to insufficient pre-training, which yields fewer predicates during HT. Overall, our method achieves acceleration ratios of 271%, 221%, and 181% on tabular, text, and image datasets, respectively. The optimization improves as the number of pre-training inputs $N$ increases. As discussed in Section 3.4, with sufficient pre-training, the average acceleration can approach the theoretical bound.

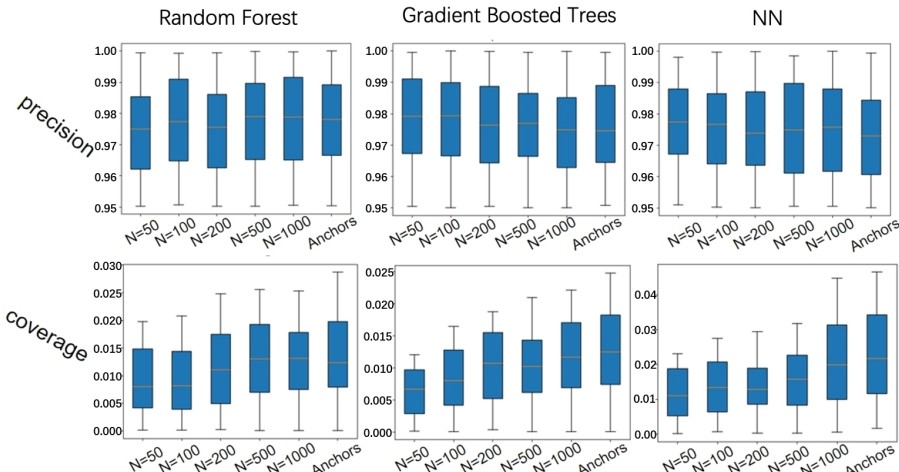

Figure 4: The fidelity of our method and Anchors on the Income Prediction task.

### 4.3 FIDELITY PRESERVATION

We use **coverage** and **precision** from Section 2.1 to evaluate the fidelity of our method. Figure 4 shows fidelity results for the income prediction task, our largest dataset. Our method reaches the target precision by iteratively adding predicates until the threshold $\tau$ is met, yielding precision nearly identical to Anchors. Coverage is slightly lower because horizontal transfer prioritizes precision over coverage, often requiring extra predicates. Increasing pre-training inputs ($N$) mitigates this by finding more similar inputs, reducing coverage loss and improving results.

## 5 RELATED WORK

Our work relates to model-agnostic explanation techniques and methods for improving their efficiency. These techniques explain model predictions without requiring access to internal details, treating models as black boxes. Prominent methods include Local Interpretable Model-agnostic Explanations (LIME) Ribeiro et al. (2016), SHapley Additive exPlanations (SHAP) Lundberg (2017), Partial Dependence Plots (PDP) Friedman (2001), Individual Conditional Expectation (ICE) plots Apley & Zhu (2020), LORE Guidotti et al. (2018), and Anchors Ribeiro et al. (2018). Among them, SHAP and Anchors achieve high fidelity and interpretability but suffer from very low computational efficiency, which is a major drawback Cummins et al. (2024).

Several efforts have been made to accelerate SHAP. TreeSHAP Lundberg (2017) speeds up SHAP for tree-based models (e.g., decision trees, XGBoost) by leveraging tree structures, reducing complexity to polynomial time while remaining exact. However, it is limited to tree-based models, greatly restricting SHAP's applicability. Accelerated Marginal Effects (AcME) Dandolo et al. (2023) is model-agnostic and approximates SHAP values for faster, scalable computation on any model, but it only works for tabular data and is unsuitable for text or image datasets. Currently, no effective acceleration method exists for Anchors.

## 6 CONCLUSION

We introduce a novel two-step rule transformation framework to significantly improve the computational efficiency of Anchors, a widely used local model-agnostic explanation technique. By leveraging pre-trained explanations and applying horizontal and vertical rule transformations, our approach adapts explanations for online inputs without sacrificing fidelity. We applied our method to various tasks on tabular, text, and image datasets. Empirical evaluations demonstrate the effectiveness of our approach in reducing the explanation time for Anchors while preserving the fidelity of the generated explanations.

## 7 ETHICS STATEMENT

This work fully adheres to the ICLR Code of Ethics. Our research does not involve human participants, does not release or use any new datasets, and raises no concerns regarding privacy, security, fairness, or potential harm. No conflicts of interest or sponsorship issues are present. All authors have read and agreed to comply with the ICLR Code of Ethics throughout the research and submission process.

## 8 REPRODUCIBILITY STATEMENT

We have taken extensive steps to ensure the reproducibility of our work. A complete conceptual description of the proposed method is provided in Section4.1 of the main paper. All datasets used are publicly available and fully cited, and the compute resources used for our experiments are described in AppendixA. An anonymous repository containing all source code, data-processing scripts, and experiment configurations is included in the supplementary materials.

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

## A    EXPERIMENTS COMPUTE RESOURCES

All the experiments was conducted on a server with 4 high-performance GPUs, each providing up to 10,000+ CUDA cores and 24GB of dedicated GPU memory, combined with 256GB system memory. The complete experiments took approximately 300 hours to run.

## B  THE USE OF LARGE LANGUAGE MODELS (LLMs)

In this paper, large language models were used solely for minor language polishing. No large language model was involved in developing the core ideas, methods, or writing of the main content.

