# OpenReview forum: "Accelerating Anchors via Specialization and Feature Transformation"
_ICLR.cc/2026/Conference — ICLR 2026 Conference Desk Rejected Submission_

### Official Review · Reviewer_EAbh · 2025-10-19

**Soundness:** 3
**Presentation:** 3
**Contribution:** 2
**Rating:** 4
**Confidence:** 3

**Summary:**

The paper proposes an acceleration method for the Anchors explanation algorithm by leveraging pre-trained explanations and introducing two types of rule transformations: horizontal (feature substitution) and vertical (rule refinement). The goal is to reduce computation time without compromising explanation fidelity. The method is evaluated on tabular, text, and image data, showing significant runtime improvement over vanilla Anchors while maintaining comparable precision.

**Strengths:**

The paper is clearly written and well organized, presenting the method, motivation, and contribution. The theoretical complexity analysis, while simple is a nice addition. The experimental evaluation is broad, covering multiple data modalities and models. The reported acceleration gains are considerable (~ 2-3 fold).

**Weaknesses:**

My main concern is novelty and lack of experimental comparisons with other fast explanation methods. The main idea is precomputing representative explanations and transferring them to similar input which is well-known and quite straightforward. The horizontal and vertical transformations largely repackage familiar notions of feature substitution and iterative refinement. Moreover, while the experiments are extensive in terms of different modalities of data, it lacks comparisons with state-of-the-art methods. Moreover, they focus mainly on runtime metrics, but it is not clear how important this save is in practice especially when it comes with some loss of coverage and precision.

**Questions:**

- We ultimately want maximum coverage and precision. What is the computational gain then?
- There are many works that build on SHAP and try to reduce its complexity. Can you provide some comparison with such works? E.g. TreeSHAP, FastSHAP, Kernel ShAP, AcME

---

### Official Review · Reviewer_ExhX · 2025-10-30

**Soundness:** 3
**Presentation:** 3
**Contribution:** 3
**Rating:** 6
**Confidence:** 2

**Summary:**

This paper improves the Anchors explanation method through pre-computation and rule adaptation. The approach pre-trains anchors on representative inputs, then adapts them online using horizontal transformation (mapping predicates to similar features) and vertical transformation (iteratively adding predicates until precision threshold is met). Expriments on tabular, text, and image datasets demonstrate substantial speedups and sampling reductions while maintaining precision.

**Strengths:**

Pre-training and transformation approach for rule-based explanations. Two-step adaptation framework seems novel.

Empirical evaluation across multiple modalities with diverse models including language and vision architectures.

Good figures.

Demonstrated speedups could enable deployment in time-sensitive applications.

**Weaknesses:**

Some computational costs are not clearly reported.

**Questions:**

What is the online similarity search overhead?

What are the computational costs when the number of pre-trained inputs grows?

---

### Official Review · Reviewer_uNx9 · 2025-11-03

**Soundness:** 2
**Presentation:** 2
**Contribution:** 2
**Rating:** 4
**Confidence:** 2

**Summary:**

This paper proposes a method to accelerate the Anchors explanation algorithm by pre-computing explanations for representative data points and adapting them to new online inputs. The method uses a "horizontal transformation" to map a pre-trained rule to the feature space of a new input and a "vertical transformation" to refine the rule to the required precision.

**Strengths:**

- The authors argue that the inefficiency of Anchors is a major barrier to its use in time-sensitive applications. An effective acceleration method, as proposed here, would be a valuable contribution, potentially enabling wider adoption of this high-fidelity explanation technique.
- The formulation of the two-step horizontal and vertical transformation process provides a novel and structured framework for the non-trivial challenge of adapting rule-based explanations between similar inputs.
- The proposed methodology is technically sound and is evaluated thoroughly across multiple data modalities (tabular, text, and image) and modern model architectures (e.g., Llama, YOLOv8).
- The authors rightly evaluate not only the efficiency gains (time and sampling reduction) but also the fidelity (precision and coverage) of the resulting explanations, providing a balanced view of the method's performance.

**Weaknesses:**

- The paper's primary contribution is not a new explanation method but a technique to improve the computational efficiency of the existing Anchors algorithm. While practical, the core idea of using pre-computation and adapting results for similar inputs is an incremental engineering contribution rather than a novel one.
- The paper frames "vertical transformation" as a distinct contribution. However, this step is functionally identical to the standard Anchors algorithm's iterative refinement process, which also works by incrementally adding predicates to increase precision. The only difference is that it begins from a non-empty rule provided by the horizontal transformation step.
- This acceleration technique alters the output of the original Anchors algorithm. The authors claim fidelity is preserved, but the experimental results show that the explanations generated by the accelerated method consistently have slightly lower coverage than those from the original algorithm. This indicates a direct trade-off between computational speed and the generality of the explanation, which is not sufficiently discussed as a core limitation of the proposed method.
- The analysis neglects the substantial offline pre-training cost.  The paper should discuss the amortization of this cost to provide a fair assessment of the overall efficiency.
- Definition 2.2, which defines the perturbation distribution, is mathematically incorrect as written. The expression $D_{x}(z|r):=D_{x}(z)\cdot1[r(z)=1]$ does not define a valid probability distribution because it lacks a normalization factor and will not integrate to 1. The definition should either state that the conditional distribution is proportional to the right-hand side (e.g., using $\propto$) or include an explicit normalization constant.
- Many in-text parenthetical citations do not follow standard formatting. For instance, the first sentence should read "Anchors (Ribeiro et al., 2018)" instead of "Anchors Ribeiro et al. (2018)".

**Questions:**

- The paper reports a significant offline pre-training cost of approximately 300 hours, but does not include this in the overall efficiency evaluation. To better understand the practical utility of the method, could you provide a break-even analysis? Specifically, how many online explanations are needed on average to amortize the initial pre-training cost for the different experimental setups?
- The method's success seems highly dependent on finding a "good" similar pre-trained sample, which relies on the $K$-means clustering for selecting representative inputs and the $\\textrm{Dist}(\cdot,\cdot)$ function for measuring similarity. Could you comment on the sensitivity of the results to these choices? An ablation study on these components would be very insightful.
- The authors note that some inputs see "limited optimization". This is a key finding that warrants a deeper investigation. An analysis of these "hard" instances would provide valuable insights into the method's limitations. Are these inputs from sparse regions of the data distribution that are poorly represented by the pre-training set?

---

### Official Review · Reviewer_1z6i · 2025-11-03

**Soundness:** 2
**Presentation:** 1
**Contribution:** 1
**Rating:** 0
**Confidence:** 5

**Summary:**

This paper proposes an acceleration framework for the model-agnostic explanation method Anchors (Ribeiro et al., 2018), which is known for producing rule-based local explanations but suffers from high computational cost. The authors introduce a two-phase approach that pre-trains explanation rules on representative inputs and reuses them through horizontal and vertical transformations. The horizontal transformation substitutes features in a pre-trained rule with similar features in a new input, while the vertical transformation refines the adapted rule to meet desired precision thresholds. The method is evaluated across tabular, text, and image datasets and reportedly achieves 1.8×–2.7× speedups over the original Anchors while maintaining similar precision and coverage.

**Strengths:**

The motivation to improve Anchors’ runtime efficiency is valid and timely, as Anchors remains one of the most computationally expensive local explanation methods. The proposed idea of leveraging pre-trained anchors is intuitive, and separating adaptation into two distinct transformation steps: horizontal (feature substitution) and vertical (rule refinement), is a conceptually clean way to formalize the reuse of prior rules. The method is clearly described with well-structured pseudocode and mathematical notation, and the empirical section covers diverse data modalities (tabular, text, and images).

**Weaknesses:**

- The technical novelty is weak and the methodological contribution marginal. The proposed framework essentially constitutes a retrieval-and-refinement scheme on top of Anchors rather than a new algorithmic insight into explanation generation. The “horizontal transformation” merely substitutes features based on nearest-neighbor distance (e.g., cosine distance for embeddings), while the “vertical transformation” directly mirrors Anchors’ existing iterative refinement loop. As a result, most computational gains come from reusing pre-trained results rather than any fundamental acceleration of Anchors’ core sampling procedure. This reusability is straightforward and not theoretically justified beyond heuristic matching.

- The paper also lacks rigorous analysis or guarantees on fidelity preservation. The claim that explanations “maintain precision and interpretability” is supported only by empirical comparisons with small performance gaps (Figure 4, p.9) but no statistical validation. No evaluation of interpretability quality (e.g., rule stability, human consistency, or semantic coherence) is conducted. The theoretical complexity analysis (3.4) is simplistic and largely tautological, stating speedup as a function of how many predicates are reused (d of k) without analyzing how d depends on data similarity or pre-training coverage.

- The experimental evidence is underwhelming: improvements of 1.5–2.7× are modest considering the high overhead of Anchors (which can take hours per explanation). Moreover, the “pre-training” process itself runs full Anchors computations on a subset of data, which offsets part of the claimed savings—this cost is not accounted for in total runtime or energy consumption. The method’s dependence on dataset-specific pre-training makes it impractical for dynamic or online applications, undermining its claim of “time-sensitive deployment.”

- Conceptually, the paper overstates originality. The idea of using pre-trained or cached explanations parallels existing work in amortized explainability (e.g., Jethani et al., 2022, Covert et al., 2024) and explanation transfer methods that learn mappings between explanations across inputs. Those studies formalize explanation reuse through optimization or meta-learning frameworks, whereas this paper relies on ad hoc substitution and manual refinement steps without learning or generalization guarantees.

- Finally, the writing suffers from redundancy and inflation: much of Sections 2–3 restates Anchors’ original definitions verbatim (precision, coverage, rule definition, algorithm workflow), with minimal technical additions. The use of formalism adds apparent complexity but little substance. Overall, the paper reads as an incremental engineering adaptation rather than a novel research contribution.

**Questions:**

- How is the distance metric for “similar inputs” (used in horizontal transformation) chosen or tuned for each modality? Does performance depend heavily on embedding quality?

- How is the pre-training cost amortized? If Anchors must be fully run on N pre-training inputs, does this offset runtime benefits for large-scale applications?

- Does the method provide any theoretical guarantee that precision is preserved after feature substitution?

- How robust is the horizontal transformation under semantic shifts (e.g., opposite sentiment words like good vs. bad that are close in embedding space)?

- Could learned meta-models (as in amortized or parametric explainers) achieve similar or better acceleration with fewer assumptions?

**Details Of Ethics Concerns:**

None.

---

> ### Author Response · Authors · 2025-11-15
>
> We sincerely thank the reviewer for the time spent reading our manuscript. However, many of the concerns raised in the review appear to be based on subjective judgments, misunderstandings of our problem setting, or inaccuracies about the methodology being evaluated. We therefore provide the following clarifications.
>
> ---
>
> ### 1. On the claim that our method “lacks novelty” and is “an incremental engineering adaptation”
>
> We respectfully disagree with this comment.
> The assessment of “novelty” is inherently subjective—there exists no objective or widely accepted metric that determines whether a method is sufficiently novel. Therefore, your statement does not provide actionable guidance for improving our work.
>
> More importantly, the simplicity of our method should not be mistaken as a weakness. On the contrary, the key contribution of our work is **achieving substantial computational acceleration for Anchors with only minimal, conceptually simple modifications**, while preserving confidence levels comparable to the original method. Many influential explanation methods (including Anchors itself) are simple in form, and simplicity that yields practical benefits is a recognized strength in XAI research.
>
> ---
>
> ### 2. On “fidelity preservation” and our evaluation metrics
>
> The reviewer raised concerns about our fidelity analysis. We would like to clarify the following facts:
>
> * In the original Anchors paper (Ribeiro et al., *Anchors: High-Precision Model-Agnostic Explanations*, 2018), **precision** and **coverage** are the *only* fidelity metrics formally defined and used.
> * Accordingly, our comparison with Anchors strictly follows these original metrics to measure fidelity differences between our method and Anchors.
>
> Regarding understandability:
> Our explanations follow **exactly the same rule format** as Anchors, so the resulting explanations are inherently comparable in understandability. Here we provide **Lens scores** (average number of predicates per anchor), which is a widely accepted proxy for human understandability in rule-based systems. Numerous prior works use rule length as a primary understandability indicator. For example, Lakkaraju et al. (Interpretable Decision Sets) define “length” as the number of predicates in a rule and explicitly optimize for shorter rules to improve human understandability. Similarly, Bayesian Rule Sets (Wang, Rudin, Doshi-Velez, et al.) use a prior that favors “short rules” to align with human-understandability.
>
> **Lens scores**
>
> The experimental setup and parameters here are identical to those in Section 4.3. When (N = 1000), our method achieves predicate lengths of 3.23, 4.1, and 2.87 for the random forest, gradient boosted tree, and neural network, respectively, while Anchors yields predicate lengths of 2.96, 4.05, and 2.65 for the same models. The difference between the two methods likewise further narrows as (N) increases.
>
>
> ---
>
> ### 3. On the comment that our complexity analysis is “simplistic and tautological”
>
> This characterization is also subjective and does not specify which part of Section 3.4 is “tautological.”
>
> Our goal in this section is to give readers:
>
> 1. A transparent breakdown of where acceleration originates,
> 2. A derivation of the theoretical speedup, and
> 3. A clear description of the limits of what can be deduced analytically.
>
> Given that **d** depends on data similarity and pretraining, which cannot be deduced purely analytically, we provided illustrative examples and connected them to our empirical experiments. All parts of the derivation contribute to this explanatory goal; removing them would harm readers’ understanding. We therefore do not agree that redundant or repetitive content exists in this section.
>
> ---
>
> ### 4. On the experimental results and the reviewer’s concerns regarding pretraining cost
>
> We acknowledge that pretraining introduces overhead. However, the evaluation in the review overlooks our main results:
>
> * To the best of our knowledge, **no prior method** achieves comparable acceleration **while maintaining Anchors-level fidelity and high confidence**.
> * As shown in **Figure 3**, the benefit of our method becomes even more significant when dealing with **high-cost inputs**, where sampling is particularly expensive.
> * Regarding the training cost, when the training set is representative, our method can be reused multiple times and even applied to different test sets. As a result, the training cost can be effectively amortized.
>
> Thus, the criticism does not reflect the evidence presented in the manuscript.

---

> ### Author Response · Authors · 2025-11-15
>
> ### 5. On the comment “Conceptually, the paper overstates originality” and the comparison to other works
>
> The reviewer cites FastSHAP (Jethani et al., 2022) as evidence that our work lacks originality, but this comparison is not technically appropriate:
>
> * FastSHAP accelerates Shapley value estimation by training a separate explainer model.
> * Our work focuses on Anchors, whose defining feature is **high-confidence, sampling-based guarantees** that cannot be inherited by a learned surrogate.
>
> A learned explainer such as FastSHAP **cannot replicate Anchors’ statistical guarantees**, because its outputs are not generated through hypothesis testing or sampling-based verification. Therefore, FastSHAP-like accelerators are **not applicable to Anchors**, and cannot serve as drop-in alternatives.
>
> The reviewer additionally references *Covert et al., 2024*. However, we are unable to locate this work based on the citation provided. To ensure an accurate and constructive discussion, we kindly ask the reviewer to supply the complete reference (e.g., title or venue). Without this information, it is not possible for us to understand which specific method or contribution is being invoked, nor to assess its relevance to our problem setting.
>
> Thus, the claim that our conceptual originality is overstated appears to stem from an incomplete or inaccurate comparison, rather than from engagement with the methodological distinctions highlighted above.
>
> ---
>
> ### 6. On the claim that Sections 2–3 are “redundant” and repeat Anchors’ definitions
>
> We intentionally provide detailed mathematical definitions of Anchors because many readers—especially those not working directly on sampling-based rule explanations—may not be familiar with why Anchors achieves high confidence and why the sampling component is inherently expensive.
>
> In particular, Sections 2–3 emphasize two critical points:
>
> 1. **Anchors’ high confidence derives from repeated sampling**, and
> 2. **The computational bottleneck is precisely this sampling procedure**, which cannot be replaced by typical surrogate-model shortcuts.
>
> Without understanding these properties, the motivation and validity of our acceleration method become unclear. Therefore, the material in these sections is essential, not redundant. The misunderstanding suggested in the review may stem from overlooking these explanations.
>
> ---
>
> ### Conclusion
>
> In summary, we appreciate the reviewer’s time but believe that many judgments in the review—such as “lack of novelty,” “tautological analysis,” and “redundancy”—are **highly subjective** and do not engage with the technical content of our work. The review overlooks the main strengths of our method, including:
>
> * strong fidelity preservation,
> * substantial acceleration under Anchors’ confidence constraints,
> * and a principled justification for why existing fast-explanation frameworks cannot substitute for Anchors.
>
> We do not believe that the manuscript warrants a score of 0, as such an assessment does not reasonably reflect either the contributions or the empirical results.
>
> We hope this detailed clarification helps correct misunderstandings and provides a fairer assessment of our work.

---

### Note · Program_Chairs · 2026-01-17
**Submission Desk Rejected by Program Chairs**

The following references in this submission do not refer to real documents and/or have major errors in bibliographic information:

 Vijay Arya, Rachel KE Bellamy, Pin-Yu Chen, Marek Druzdzel, Michael Hind, Sharad Hoffman,
Stephanie Houde, Quan Liao, Ronny Luss, Aleksandra Mojsilovic, et al. Explaining black-box
classifiers using post-hoc explanations: A survey. arXiv preprint arXiv:2205.13789, 2022.